# Design and Validation of a Simplified Method to Determine Minimum Bactericidal Concentration in Nontuberculous Mycobacteria

**DOI:** 10.3390/antibiotics14040381

**Published:** 2025-04-04

**Authors:** Sara Batista, Mariana Fernández-Pittol, Lorena San Nicolás, Diego Martínez, Sofía Narváez, Mateu Espasa, Elena Garcia Losilla, Marc Rubio, Montserrat Garrigo, Griselda Tudó, Julian González-Martin

**Affiliations:** 1Unitat de Microbiologia, Departament de Fonaments Clínics, Facultat de Medicina i Ciències de la Salut, Universitat de Barcelona, c/Casanova 143, 08036 Barcelona, Spain; sbatista@recerca.clinic.cat (S.B.); mjfernandez@clinic.cat (M.F.-P.); mespasa@clinic.cat (M.E.); 2Fundació de Recerca Clínic Barcelona–Institut d’Investigacions Biomèdiques Augustí Pi i Sunyer (FRCB-IDIBAPS), 08036 Barcelona, Spain; 3Servei de Microbiologia, CDB, Hospital Clínic de Barcelona, c/Villarroel 170, 08036 Barcelona, Spain; lsannic@clinic.cat (L.S.N.); dmartine@clinic.cat (D.M.); 4IsGlobal Barcelona, Institute for Global Health, 08036 Barcelona, Spain; 5Catlab, Centre Analítiques Terrassa AIE, Servei de Microbiologia Vallès Occidental, Parc Logístic de Salut, Viladecavalls, 08232 Terrassa, Spain; sanarvaez@catlab.cat; 6Servei de Microbiologia, Fundació de Gestió de l’Hospital de la Santa Creu i Sant Pau, 08025 Barcelona, Spain; egarcia@santpau.cat (E.G.L.); mrubiobu@santpau.cat (M.R.); mgarrigo@santpau.cat (M.G.); 7Institut d’Investigació Biomèdica Sant Pau (IIB SANT PAU), c/Sant Quintí, 89, 08026 Barcelona, Spain; 8Sant Pau Institute for Biomedical Research, 08041 Barcelona, Spain; 9CIBER of Infectious Diseases (CIBERINFEC), Instituto de Salud Carlos III, 28029 Madrid, Spain

**Keywords:** nontuberculous mycobacteria (NTM), reincubation method, subculturing method, SLOMYCOI Sensititre, antibiotics, minimum bactericidal concentration (MBC), bacteriostatic/bactericidal activity

## Abstract

Background/Objectives: Nontuberculous mycobacteria (NTM) infections are rising, particularly those by *Mycobacterium avium* complex (MAC) and *Mycobacterium abscessus* complex (MAB). Treating NTM infections is challenging due to their poor response to antibiotics. This study aimed to optimize the treatment of NTM infection by selecting antibiotics with bactericidal activity for combination therapy. To do this, we used the minimum bactericidal concentration (MBC) determination approach to define bactericidal or bacteriostatic activity. We developed three main objectives: validate a new method to determine MBC based on a reincubation method, determine MBC values of 229 NTM clinical isolates using the reincubation method, and evaluate antibiotic stability in preincubated microtiter plates. Methods: First, we assessed the stability of the antibiotics included in SLOWMYCOI Sensititre™ microtiter plates. Five strains of MAC were studied comparing the minimum inhibitory concentrations (MICs) of those preincubated for seven days vs. non-incubated plates. Then, we evaluated the percentage of reproducibility of MBC values using two methods, reincubation and subculturing (standard or traditional method) in 30 MAC isolates. Finally, we validated the reincubation method and prospectively determined the MBC values of the 229 NTM clinical strains. Results: Antibiotic stability: The MIC was equivalent after 7 and 14 days for all the antibiotics, except rifampicin, for which the MIC increased by 2- to 3-fold after preincubation. Reincubation method: The percentage of reproducibility of the MBC values between the two methods was 95.2% (range 76.6% to 100%). Prospective validation: MBC/MIC ratios revealed differential bactericidal activity for most antibiotics according to the different species, being bactericidal in *M. avium* and *Mycobacterium xenopi*, and predominantly bacteriostatic in MAB. Conclusions: Preincubation of Sensititre™ microtiter plates did not alter the MIC values of the antibiotics included except for rifampicin, suggesting a loss of activity. MBC determination can be easily performed by the Reincubation method presented. MBC values provide useful additional information regarding MIC values since the MBC/MIC ratio reveals whether antibiotics have bactericidal or bacteriostatic activity according to the species, which is pivotal for selecting the most adequate antibiotic combination to ensure efficient treatment management.

## 1. Introduction

Mycobacterial species share unique features: they are acid-alcohol resistant, their cell walls have a lipid layer of mycolic acids, and they grow very slowly [1], with a growth rate of between 6 and 24 h, depending on the species. These attributes endow mycobacteria enhanced resilience to adverse conditions, disinfectants, and antibiotics. Despite *Mycobacterium tuberculosis* being the primary causative agent of global infections, there has been a notable increase in infections caused by nontuberculous mycobacteria (NTM) [2]. Among these, infections attributed to *Mycobacterium avium* complex (MAC) and *Mycobacterium abscessus* complex (MAB) are the most prevalent [3]. NTM originate from environmental sources, such as soils, plants, and water [4] resulting in widespread exposure within the general population, often leading to colonization without symptomatic manifestations of the disease. MAC infection is often correlated with the immune status of individuals [5]. The most common presentations of MAC infections involve the lungs, notably affecting patients with underlying chronic respiratory conditions, such as chronic obstructive pulmonary disease, bronchiectasis, and cystic fibrosis [6]. NTM are inherently resistant to many antibiotics due to their lipid-rich cell wall, slow growth rate, and ability to form biofilms, which reduce drug penetration and efficacy. Therefore, the management, clinical diagnosis, and the treatment of NTM lung infections in patients with chronic respiratory disease are complex [7] due to poor response to antibiotic regimens and the tendency to chronicity in these patients. Currently, there is a lack of standardized treatment for NTM, although there are empirical recommendations [7] especially for MAC and MAB, which are the most prevalent causes of pulmonary infection. Treatment typically entails prolonged combined antibiotic regimens including macrolides, amikacin, tigecycline, and others. The therapy, which lasts 12 months or more, usually includes 3–4 antibiotics with different mechanisms of action [6,8], to prevent the development of resistance and achieve successful eradication. The long duration and the multidrug combination treatment may incur significant toxicity and side effects, thereby potentially leading to treatment discontinuation and treatment failure.

Knowledge of the bactericidal activity of antibiotics used in NTM treatment is important for optimizing treatment regimens [9]. Minimum bactericidal concentrations (MBCs) measure the bactericidal activity of antibiotics and are an indicator of drug effectiveness, being useful to design the most effective drug combinations to shorten treatment duration and achieve better outcomes and prognoses. A technique that is efficient, rapid, practical, cost-effective, and suitable for routine laboratory use should be used to determine MBCs. However, the absence of an optimized technique for MBC determination remains a challenge. The standard (traditional method) relies on determining the minimum inhibitory concentration (MIC) by microdilution and subsequently plating the contents of the MIC wells onto solid agar to assess bacterial viability by quantifying colony-forming units (CFUs), and this process is laborious and tedious. According to this method, the MIC of the antibiotic inhibits 99% of the bacterial population, and the MBC inhibits 99.99% [10]. While the subculturing method is one approach for MBC determination, some other modified versions of the reincubation, subculturing, and CFU counting methods are being used [10] (Appendix A, Appendix A). Nevertheless, there is a need to develop a simplified technique for determining the MBC to aid treatment decision-making in patients with NTM infections. Additionally, the long incubation period to determine MIC and MBC values, particularly in slow-growing mycobacteria (SGM), can contribute to antibiotic degradation and subsequent loss of activity, which can lead to MIC misinterpretation. As described in previous studies [11,12], the stability of antibiotics may be altered over time, and thus, we first verified if the MBC determination method of reincubation, developed by our group, could also modify antibiotic stability. The reincubation method involved incubating the MIC plate for a second period (3 days for rapid-growing mycobacteria (RGM), 7 days for SGM) without modifying any conditions, and then reading the plate by visualizing bacteria growth in a visual plate reader. We used commercial 96-well plates (Sensititre microtiter plates) that contained predefined lyophilized antibiotic concentrations that are commonly used for treating mycobacterial infections. The advantage of these plates is that they obviate the need for manual antibiotic preparation, reducing experimental error. Additionally, they simplify the process, as only the inoculum and culture medium need to be added to the wells to determine the MIC.

The purpose of this study was to evaluate a new method to determine the MBC in NTM clinical isolates by reincubating the MIC plates for a second period of time and then comparing the MBC values obtained with those of the standard method following the microdilution technique using commercial 96-well microtiter plates, also called the flash microbiocide method [13]. This technique has been widely used since 2013 to determine the MBC in various bacterial species [14,15,16,17,18,19,20,21,22,23,24,25], including mycobacteria [26].

## 2. Results

### 2.1. Antibiotic Stability

The preincubation of SLOMYCOI Sensititre™ microtiter plates did not alter the MIC values of the antibiotics after more than 2 dilutions except for rifampicin (RIF). In 3 out of 5 clinical isolates, the MIC values increased by more than two dilutions for RIF when preincubated for 7 days at 37 °C (Table 1).

### 2.2. Pilot Assay

The percentage of the reproducibility of MBC values between the two methodologies tested (the standard and the reincubation method) to determine MBC values is shown in Figure 1, ranging from 76.6% for ethionamide (ETH) to 100% for amikacin (AMK), clarithromycin (CLA), linezolid (LIN), and ethambutol (EMB). ETH showed the greatest variation in MBC values with both methodologies (from 0.3 to 20 µg/mL) (Appendix A).

### 2.3. Implementation of the Reincubation Method

MBC values were calculated using the reincubation method for each of the six most frequently isolated NTM species in our area. The MIC90 and MBC90 values of the SGM are shown in Figure 2. The antibiotics that were active against *M. intracellulare-chimaera* (N = 99) were AMK, CLA, moxifloxacin (MOX), and rifabutin (RFB). Among these, RFB had an MBC/MIC ratio of 4, indicating bactericidal activity. LIN was not only active against this species but also revealed a bactericidal effect with an MBC/MIC ratio of 2. AMK. CLA and RFB were active against *M. avium* (N = 64) isolates with an MBC/MIC ratio < 4, displaying bactericidal activity. Isolates of *M. xenopi* (N = 17) were susceptible to AMK, ciprofloxacin (CIP), CLA, and LIN with an MBC/MIC ratio of 1, thus showing a bactericidal effect. Streptomycin (STR), cotrimoxazole (SXT), and RIF had an MBC/MIC ratio of 2, exhibiting bactericidal activity.

The MIC90 and MBC90 values of the RGM are shown in Figure 3. MAB isolates (N = 29) were the most resistant species, being susceptible to two of the most frequently used antibiotics to treat this species: AMK and tigecycline (TGC). However, the MBC/MIC ratios were >4 for both antibiotics, demonstrating bacteriostatic activity. Likewise, in *M. chelonae* (N = 13), the MBC/MIC ratio of TGC as well as doxycycline (DOX) and minocycline (MIN) was >4. CLA and tobramycin (TBR) had an MBC/MIC ratio <4, displaying bactericidal activity. *M. fortuitum* (N = 7) isolates were the least resistant species among the three tested in this study, being susceptible to TGC, AMK, CIP, and MOX. The MBC/MIC ratio was >4 for AMK, CIP, imipenem (IMI), LIN, and MOX except for TGC, for which it was 2, implying bactericidal activity.

For both SGM and RGM, the MBC values for the antibiotics not mentioned above could not be determined with the exact increase in the number of dilutions because the MBC values surpassed the highest tested concentration (Appendix A).

## 3. Discussion

Establishing the most adequate treatment for patients with an NTM infection is a major goal. Determination of the MBC can be useful for clinical decision-making in the treatment of NTM infections. Some antibiotics have suitable MICs but do not exhibit bactericidal behavior. In order to select the most adequate antibiotic, not only should antibiotics showing an acceptable MIC be considered, but the bactericidal activity, which is determined by the MBC, should also be taken into account. In this sense, the aim of this study was to design an easier and reproducible methodology to determine the MBC in order to identify which antibiotics have bactericidal activity against six of the most frequently isolated NTM species in our area. The study was performed using a microdilution method with the commercial microtiter Sensititre™ RAPMYCOI/SLOMYCOI plates. The microdilution method for MBC determination employed in this study is a well-established and widely utilized technique. It has been applied to various bacterial species [14,15,16,17,18,19,20,21,22,23,24,25], including mycobacteria [26]. Compared to the CFU-based method, microdilution is less time- and labor-intensive, as it eliminates the need for agar plating. Instead, it requires only the transfer of cultures into fresh liquid media in a microtiter plate. Furthermore, this approach reduces costs by approximately 60% compared to the CFU-based method [13]. To achieve this objective, we had to first confirm the stability of the antibiotics during the second period of incubation by preincubating Sensititre™ plates. The pilot assay of the preincubation of the commercial plates revealed the same MIC values for the antibiotics tested as those that were not preincubated, except for RIF. This antibiotic did not maintain the same activity after 14 days, with the MIC values being 2- or 3-fold higher than those not preincubated. Previously published studies [11,12] have reported that some antibiotics, including RIF, degrade after 14 days. We found that only RIF showed this phenomenon after one week of preincubation. It was of note that RFB, which is included in the same antibiotic family, showed no variation in MIC values after preincubation. The loss of RIF activity over time compromises the reliability and interpretability of MBC determination. Therefore, MBC determination of RIF by the reincubation method was not performed in this study. Compound-specific factors such as solubility, stability, and interaction with media components might also affect the reincubation method for the MBC determination. Therefore, prior MBC testing by comparing reincubation and subculturing methods should be conducted in other or novel drugs that have not been tested in this study.

Reincubation of MIC plates for a second period of time resulted in MBC values like those obtained with the standard methodology. The percentage of reproducibility of MBC values was between 90–100% for AMK, CLA, EMB, LIN, and MOX, and 76% for ETH. Thus, we considered the reincubation method to be an efficient and reliable procedure to determine the MBC. The reincubation method reduces the time required while minimizing the risk of contamination and eliminates the need for extra reagents and materials.

After the validation of the pilot assay, we used this method to determine the MBC values of 229 NTM obtained over a period of 18 months. MBC values provide additional information since the MBC/MIC ratio represents the bactericidal activity of an antibiotic. This ratio can provide information for selecting appropriate antibiotic combinations to optimize treatment strategies for NTM infections. However, determining the in vitro MBC/MIC ratio may influence the measurements of antibiotic bactericidal activity due to various biological and technical factors. These factors include tolerance, phenotypic resistance, paradoxical effect, persisters, the growth phase of the inoculum, the volume transferred, antibiotic carryover, and the composition of the culture medium [27]. In this study, the bactericidal activity was found to be species-dependent for each antibiotic. For instance, MAB was only susceptible to TGC and AMK, but both antibiotics had a bacteriostatic effect on this species. Despite having extremely low MICs, these two antibiotics do not seem to be the most suitable choices. Conversely, in *M. fortuitum*, TGC had bactericidal activity. Although TGC typically presents low MIC values, it is associated with poor tolerability and limited clinical efficacy in patients [28]. In the case of SGM, AMK and CLA were the only antibiotics active against *M. intracellulare* complex clinical isolates, but they also had bacteriostatic activity, whereas for *M. xenopi* most of the antibiotics tested had bactericidal activity and were susceptible to most of the antibiotics included in the microtiter plates.

In parallel, the present study detected that some of the antibiotics included in the Sensititre™ microtiter plates are not adequate for some *Mycobacterium* species, especially beta-lactams and tetracyclines. Indeed, NTM produces beta-lactamases, and although there are few reported cases of NTM not producing these enzymes [29,30], it does not seem reasonable to include beta-lactams in the plates. Several antibiotics included in the Sensititre™ plates, such as amoxicillin clavulanic acid (AUG), ceftriaxone (AXO), DOX, cefepime (FEP), cefoxitin (FOX), and SXT, did not show activity against MAB, *M. chelonae*, or some *M. fortuitum* isolates. Likewise, for SGM species, CIP, DOX, STR, and SXT showed no activity against *M. avium*, *M. intracellulare* complex, and *M. xenopi*. We suggest that these antibiotics should be reconsidered by the manufacturer. Other antibiotics could be candidates, such as clofazimine, bedaquiline, and others recently reported, such as new tetracyclines, oxazolydines, and glycopeptides [31].

The key findings of this study are as follows: (1) The MBC using the reincubation method yielded results comparable to the subculturing method, establishing reincubation as a reliable, efficient, and easier method for MBC determination in NTM isolates. (2) RIF loses activity after 14 days of incubation at 37 °C, and this may lead to misinterpretation of the MIC (Table 1). (3) The MBC determination of 229 NTM revealed that the bactericidal or bacteriostatic activity of each antibiotic is species-dependent. (4) Most antibiotics included in Sensititre RAPMYCOI and SLOWMYCOI commercial plates are not active against NTM.

## 4. Materials and Methods

### 4.1. Antibiotic Stability

#### 4.1.1. Isolate Selection

A total of five clinical MAC isolates were selected based on their low MIC profile to determine antibiotic stability. All five isolates were respiratory samples collected from patients with chronic pulmonary disease analyzed in the Microbiology Department of the Hospital Clinic of Barcelona, Spain.

#### 4.1.2. MIC Determination

The determination of MICs was performed using a widely adopted technique with commercial 96-well microtiter plates containing lyophilized antibiotics at known concentrations. A total of 100 µL/well of Middlebrook 7H9 liquid medium was added to five SLOMYCOI Sensititre™ Thermo Scientific™ microtiter plates (Thermo Fisher Scientific, Waltham, MA, USA), and these were then sealed and preincubated for 1 week at 37 °C. After 1 week, 100 µL of inoculum adjusted to 3 × 10^5^ CFU/mL was added to each well of the 5 plates. The same day, 100 µL of Middlebrook 7H9 fresh medium followed by 100 µL of the same inoculum adjusted to 3 × 10^5^ CFU/mL were also added to 5 new plates that had not previously been incubated. Each well of the 10 plates contained a total volume of 200 µL. The plates were sealed and incubated for 7 days and read in a plate reader VIZION System (SWIN^®^ version 3.3.2.7). The experiment was carried out in duplicate.

### 4.2. The New Reincubation Method to Evaluate MBC Determination: Pilot Assay

#### 4.2.1. Isolate Selection

A total number of 30 clinical MAC isolates were obtained over two months to determine the MBC of standard subculturing versus the new reincubation method. The 30 isolates were respiratory samples collected from patients with chronic pulmonary disease analyzed in the Microbiology Department of the Hospital Clinic of Barcelona.

#### 4.2.2. MIC Determination

The MICs of 30 MAC clinical isolates were determined with the seven most active and frequently used antibiotics to treat MAC infections: AMK, CLA, RFB, MOX, ETH, EMB, and LIN. RIF was not included due to the lack of isolates with MIC and MBC values below the highest concentration available in Sensititre™ microtiter plates. The microdilution method was performed using SLOMYCOI Sensititre™ microtiter plates following the manufacturer’s instructions. The MICs were determined using the reader VIZION system. The experiment was carried out in duplicate.

#### 4.2.3. MBC Determination by the Standard Method

An adapted flash microbiocide method [13,26,32] was used to determine the MBC. Briefly, once the MIC had been determined as described in the MIC determination section, plate wells containing the above-mentioned antibiotics were subcultured transferring 20 µL/well to a new 96-well plate containing 180 µL of fresh Middlebrook 7H9 liquid culture and incubated for 1 week at 37 °C. The first well without visible growth was considered the MBC. The experiment was carried out in duplicate.

#### 4.2.4. MBC Determination by the Reincubation Method

In parallel, the original MIC plates were incubated for one more week. Wells containing the seven antibiotics were read again using the reader VIZION system. The first well without visible growth was considered the MBC. Result interpretation: MBC values obtained by the two methods were compared by calculating the percentage of reproducibility of the MBC values. The percentage of reproducibility of the MBC values between the reincubation and subculturing method was calculated as follows: % of reproducibility of the MBC values = (number of isolates with the same MBC using both the subculturing and reincubation method/30 total tested isolates) × 100. If the MBC values of the two methods were the same or differed by ≤1 dilution, the MBC value was considered as the same. The experiment was carried out in duplicate.

### 4.3. Implementation of the Reincubation Method

#### 4.3.1. Isolate Recruitment

A total of 229 clinical NTM isolates were prospectively collected over a period of 18 months to determine the MBC using the reincubation method. Among the 229 isolates, 49 were RGM, and 180 were SGM with 29 *M. abscessus*; 13 *Mycobacterium chelonae; and* 7 *Mycobacterium fortuitum*, and 163 were MAC isolates including 99 *M. intracellulare-chimaera*; 64 *M. avium*; and 17 *M. xenopi*. The isolate selection criteria were defined as follows: species for which at least 7 strains were isolated and whose MIC values were 2 dilutions lower than the highest tested antibiotic concentration and 1 dilution higher than the lowest tested antibiotic concentration. A total of 13 antibiotics were tested in SGM and 15 antibiotics in RGM. For each antibiotic, the number of isolates analyzed was considered according to the number of isolates with MIC values between the ranges of each antibiotic in the Sensititre™ panel. All of the 229 isolates were respiratory samples collected from patients with chronic pulmonary disease analyzed in the Microbiology Department of the Hospital Clinic of Barcelona.

#### 4.3.2. MIC Determination

The MIC was determined by performing the microdilution method (RAPMYCOI or SLOMYCOI Sensititre™) following the manufacturer’s instructions as described in the MIC determination section and following Clinical and Laboratory Standards Institute guidelines [33]. For RGM, the following antibiotics were included in the RAPMYCOI Sensititre™ plates: STX, LIN, CIP, IMI, MOX, FEP, FOX, AUG, AMK, AXO, DOX, MIN, TGC, and TBR. The liquid medium culture of cation-adjusted Mueller–Hinton broth was used for RGM.

For SGM, the following antibiotics were included in the SLOWMYCOI Sensititre™ microtiter plates: CLA, CIP, STR, DOX, ETH, RFB, STX, EMB, isoniazid (INH), MOX, RIF, AMK, and LIN. The Middlebrook 7H9 liquid medium supplemented with 10% oleic acid-albumin-dextrose-catalase was used in SGM.

#### 4.3.3. MBC Determination by the Reincubation Method

Following MIC determination, the plates were incubated for one more week. Wells containing the antibiotics were read again using the reader VIZION system. The first well without visible growth was considered the MBC. MICs_90_ and MBCs_90_ were calculated for each species. MIC_90_ and MBC_90_ refer to the MIC and MBC values at which 90% of the isolates were inhibited or killed, respectively. Result interpretation: If the MBC_90_/MIC_90_ ratio was ≤4, the antibiotic was considered to have bactericidal activity, while for bacteriostatic activity, the ratio was >4 [34,35,36]. The experiment was carried out in duplicate.

## 5. Conclusions

The main findings of this study are that the reincubation method for the MBC determination yielded results comparable to the standard subculturing method, establishing it as a reliable, efficient, and practical alternative.

Preincubation of Sensititre™ microtiter plates did not affect the MIC values of the tested antibiotics, except for RIF, which exhibited a loss of activity after 14 days of incubation at 37 °C, potentially leading to MIC misinterpretation.

MBC determination of 229 NTM isolates confirmed that the bactericidal or bacteriostatic activity of each antibiotic is species-dependent.

Most antibiotics included in the Sensititre RAPMYCOI and SLOWMYCOI plates showed limited activity against NTM.

## Figures and Tables

**Figure 1 antibiotics-14-00381-f001:**
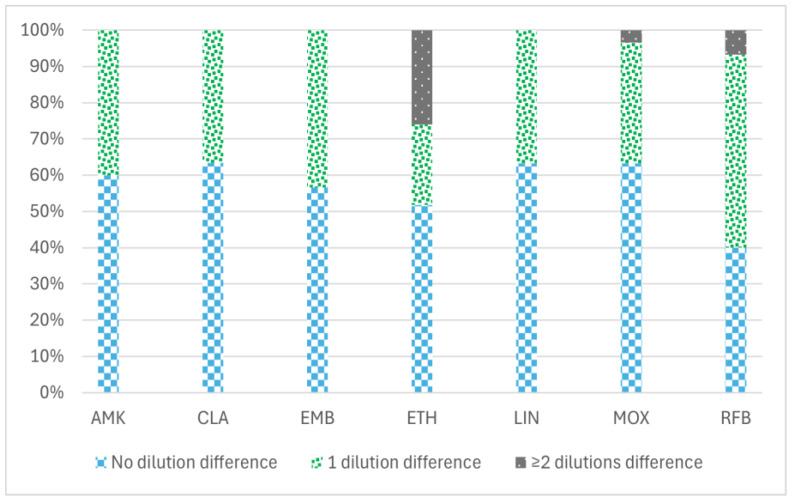
Percentage of reproducibility of MBC values between two different minimum bactericidal concentration (MBC) determination methodologies in 30 MAC isolates with seven antimicrobial agents: amikacin, clarithromycin, ethambutol, ethionamide, linezolid, moxifloxacin, and rifabutin. The data are categorized into three groups based on the variation in MBC values obtained by the two methods (standard and reincubation): the percentage of isolates demonstrating no differences (0 dilution), the percentage exhibiting a difference of one dilution (1 dilution), and the percentage displaying two or more dilution differences (≥2 dilutions) between the MBC values derived from each method. AMK: amikacin, CLA: clarithromycin, EMB: ethambutol, ETH: ethionamide, LIN: linezolid, MOX: moxifloxacin, and RFB: rifabutin.

**Figure 2 antibiotics-14-00381-f002:**
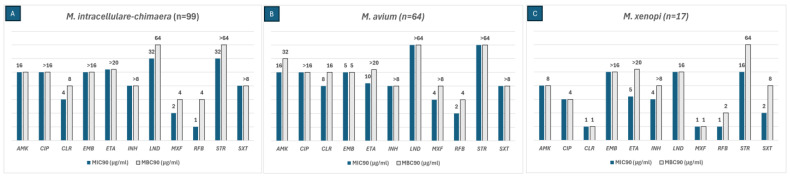
The MIC_90_ (blue bars) and MBC_90_ (grey bars) values of three SGM species for the antibiotics included in the SLOMYCOI plate.The scale of the vertical axis has been adapted to better represent the MBC and MIC dilution distances. MIC: minimum inhibitory concentration, MBC: minimum bactericidal concentration. AMK: amikacin, CIP: ciprofloxacin, CLA: clarithromycin, EMB: Ethambutol, ETH: ethionamide, INH: isoniazid, LIN: linezolid, MOX: moxifloxacin, RFB: rifabutin, STR: streptomycin, and SXT: cotrimoxazole. Antibiotic (total number of isolates analyzed for each antibiotic only included the isolates with an MIC between the range of each antibiotic in the Sensititre™ panel); (**A**) AMK (98), CIP (86), CLA (98), EMB (87), ETH (57), INH (63), LIN (94), MOX (99), RFB (98), STR (81), SXT (37); (**B**) AMK (64), CIP (55), CLA (62), EMB (58), ETH (60), INH (48), LIN (62), MOX (63), RFB (57), STR (53), SXT (29); (**C**) AMK (17), CIP (17), CLA (14), EMB (12), ETH (15), INH (15), LIN (17), MOX (16), RFB (13), STR (17), SXT (11).

**Figure 3 antibiotics-14-00381-f003:**
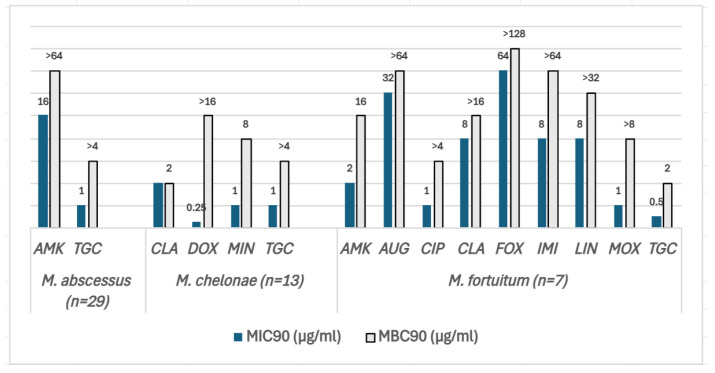
The MIC_90_ (blue bars) and MBC_90_ (grey bars) values of three SGM species for the antibiotics included in the SLOMYCOI plate. The scale of the vertical axis has been adapted to better represent the MBC and MIC dilution distances. MIC: minimum inhibitory concentration, MBC: minimum bactericidal concentration. AMK: amikacin, AUG: amoxicillin clavulanic acid, CIP: ciprofloxacin, CLA: clarithromycin, DOX: doxycycline, FOX: cefoxitin, IMI: imipenem, LIN: linezolid, MIN: minocycline, MOX: moxifloxacin, and TGC: tigecycline. Antibiotic (total number of isolates analyzed for each antibiotic only included the isolates with an MIC between the range of each antibiotic in the Sensititre™ panel). *M. abscessus* (N = 29): AMK (27), TGC (29); *M.chelonae* (N = 13): CLA (13), DOX (2), MIN (1), TGC (13); *M. fortuitum* (N = 7): AMK (7), AUG (6), CIP (7), CLA (7), FOX (7), IMI (7), LIN (7), MOX (7), and TGC (7).

**Table 1 antibiotics-14-00381-t001:** The MICs of five *M. intracellulare* complex clinical isolates with the antibiotics included in SLOMYCOI plates after prior incubation.

MIC (µg/mL) *M. intracellulare* Complex										
	Isolate 1		Isolate 2		Isolate 3		Isolate 4		Isolate 5	
	Standard Protocol *	Preincubation	Standard Protocol	Preincubation	Standard Protocol	Preincubation	Standard Protocol	Preincubation	Standard Protocol	Preincubation
AMK	8	8	4	8	2	8	16	32	16	16
CIP	8	8	2	2	1	1	8	8	8	8
CLA	4	8	1	1	1	2	2	2	2	2
DOX	>8	>8	>8	>8	>8	>8	>8	>8	>8	>8
EMB	2	2	1	2	2	2	4	4	4	4
ETH	10	10	>10	>10	1.25	1.25	1.25	1.25	>10	>10
INH	4	4	>4	>4	4	4	2	2	>4	>4
LIN	16	16	8	16	16	32	16	16	16	16
MOX	0.5	0.5	0.5	0.5	1	1	0.5	0.5	1	1
RFB	0.25	0.5	0.125	0.25	0.125	0.25	0.125	0.25	0.25	0.125
RIF	0.5	1	0.5	2	0.25	4	1	4	1	>4
STR	8	8	4	4	8	8	8	16	16	16
SXT	>4	>4	>4	>4	>4	>4	2	2	>4	>4

MIC: Minimum inhibitory concentration, AMK: amikacin, CIP: ciprofloxacin, CLA: clarithromycin, DOX: doxycycline, EMB: Ethambutol, ETH: ethionamide, INH: isoniazid, LIN: linezolid, MOX: moxifloxacin, RFB: rifabutin, RIF: rifampicin, STR: streptomycin, and SXT: cotrimoxazole. * Standard protocol: non-previously incubated SLOMYCOI plates, 37 °C days: preincubation: 7 days preincubated plates at 37 °C.

## Data Availability

The original contributions presented in this study are included in the article/Appendix A. Further inquiries can be directed to the corresponding author(s).

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
