# Peer review of "Design and Validation of a Simplified Method to Determine Minimum Bactericidal Concentration in Nontuberculous Mycobacteria"

_antibiotics, 2025, doi:10.3390/antibiotics14040381_

Round 1

Reviewer 1 Report

Comments and Suggestions for Authors

The study provides a new method for determining MBC concentrations of various antibiotics agains non-MTb Mycobacteria. The method is clearly described and scientifically sound and can be useful for both research and clinical labs.

The manuscript discusses antibiotic efficacy, but does not really discuss resistance. While the lack of activity of some classes and specific antibiotics is discussed, the discussion would benefit from a paragraph highlighting the challenges of treating resistant strains and the potential of the method to screen for resistance quickly, therefore potentially providing clinical guidance for treatment options.

The manuscript is missing a comment on the antibiotic activity based on nutritional availability (i.e. growth medium used). Depending on the drug the activity may be worse or better depending on the media used, and the study only used 7H9. Conditions in the human body will also vary widely, as such efficacy depending on nutrient availability is an important factor to consider.

Introduction should expand on some of the challenges of treating Mycobacterial infections as many of these are not shared with other bacterial species. Treatment options should be discussed, e.g. why multiple drugs must be used, treatment lengths, stress on the patient, etc. Can synergistic combinations be used? Can this method test for synergistic or other combinatorial MBCs?

Comments on the Quality of English Language

Please add spaces between numbers and units/symbols throughout the manuscript. All temperatures are missing a space between the number and the unit. Line 147 and Table 1 missing spaces between numbers and symbols.

Either use Mycobacterial species or Mycobacteria, not Mycobacteria species.

Be consistent with grammar. Either use the Oxford comma (my preference) or don't, but don't sometimes use it and sometimes not use it. E.g. lines 62 and 72 compared to lines 67 and 124.

Line 86: clarify sentence.

Line 214 remove comma.

Line 203 - remove "Interestingly" from the start of the sentence. It is used again on line 205.

Author Response

Comment 1: The manuscript discusses antibiotic efficacy but does not really discuss resistance. While the lack of activity of some classes and specific antibiotics is discussed, the discussion would benefit from a paragraph highlighting the challenges of treating resistant strains and the potential of the method to screen for resistance quickly, therefore potentially providing clinical guidance for treatment options. 

Response 1: We appreciate these insightful comments. The aim of the study was not to assess resistance but rather to highlight the valuable information provided by MBC determination regarding bactericidal activity, helping to identify the most potent antibiotics for selecting combination treatment. While our manuscript primarily focuses on antibiotic efficacy, we acknowledge the importance of discussing resistance, particularly in the context of treatment challenges and the potential of our method to provide rapid clinical guidance. We have highlighted the challenges of treating resistant strains in the Introduction section, page 2-3, paragraph 1, lines 75-78. 

Comment 2: The manuscript is missing a comment on the antibiotic activity based on nutritional availability (i.e. growth medium used). Depending on the drug the activity may be worse or better depending on the media used, and the study only used 7H9. Conditions in the human body will also vary widely, as such efficacy depending on nutrient availability is an important factor to consider. 

Response 2: Thank you for your comment. Indeed, some NTM species have specific nutrition requirements. We have now provided details of the medium culture used for each rapid and slow growing mycobacteria species in the Method section (page 10, paragraphs 2 and 3, lines 356-357, 360-361). We are aware of the limitation of the in vitro experiments, and we have given some examples of the biological and technical factors that might interfere in vitro measurement of the bactericidal activity of the antibiotics in the Discussion section as a limitation of the study (page 7, paragraph 3, lines 245-249).   

Comment 3: Introduction should expand on some of the challenges of treating Mycobacterial infections as many of these are not shared with other bacterial species. Treatment options should be discussed, e.g. why multiple drugs must be used, treatment lengths, stress on the patient, etc. Can synergistic combinations be used? Can this method test for synergistic or other combinatorial MBCs? 

Response 3: We also recognize that this is a major issue and acknowledge that we could have placed greater emphasis on it. Therefore, we have expanded on the challenges of the treatment by adding more details in a paragraph included in the Introduction (page 2, paragraph 3, page 3 paragraph 1, lines 75–78 and 82–88, respectively). 

Regarding synergistic combinations, MBC is useful for assessing bactericidal activity but not for evaluating synergy as the reviewer suggests. 

Comment 4: Comments on the Quality of English Language 

Please add spaces between numbers and units/symbols throughout the manuscript. All temperatures are missing a space between the number and the unit. Line 147 and Table 1 missing spaces between numbers and symbols.  

Either use Mycobacterial species or Mycobacteria, not Mycobacteria species.  

Be consistent with grammar. Either use the Oxford comma (my preference) or don't, but don't sometimes use it and sometimes not use it. E.g. lines 62 and 72 compared to lines 67 and 124. 

Line 86: clarify sentence.  

Line 214 remove comma.  

Line 203 - remove "Interestingly" from the start of the sentence. It is used again on line 205.  

Response 4: We agree and are pleased to incorporate all the English language corrections into the manuscript. It has been thoroughly revised by a native English language expert before and after the resubmitted version of the manuscript. 

Reviewer 2 Report

Comments and Suggestions for Authors

Review for Batista et al for the manuscript “Design and validation of a simplified method to determine minimum bactericidal concentration in nontuberculous myco-bacteria”

This study focuses on developing and validating a Reincubation Method for determining Minimum Bactericidal Concentration (MBC) in nontuberculous mycobacteria (NTM), particularly Mycobacterium avium complex (MAC) and Mycobacterium abscessus complex (MAB). Given the challenges in treating NTM infections, accurately distinguishing between bactericidal and bacteriostatic antibiotics is essential.

The study compared the new method to the traditional subculturing method and assessed the stability of antibiotics in preincubated Sensititre™ microtiter plates. Results indicated that the Reincubation Method correlates well (95.2%) with the standard technique, making it a simpler and more efficient alternative. Antibiotic stability tests showed that most antibiotics remained stable over time, except for rifampicin, which exhibited a loss of activity after preincubation. The MBC/MIC ratio analysis demonstrated that antibiotics were bactericidal against M. avium and M. xenopi but mostly bacteriostatic against MAB

However, the study lacks detailed statistical analysis, such as confidence intervals, p-values, and comparative tests, which are important to validate the reliability of the new method.

Major concerns-

I have a major issue with the RESULTS section of this manuscript as the authors JUMP directly to the conclusion without providing any context or introduction. This makes it difficult for readers to understand the significance of the results or how they relate to the study’s objectives

Please describe what is Reincubation method in the introduction for general audience.

What is so special about these SLOMYCOI Sensititre™ microtiter plates? Please describe briefly for the journal audience. No explanation of why these plates were chosen, their specific composition, or their advantages. Please provide relevant information in introduction/result section for general audience.

Line 43- The manuscript states a 95.2% correlation but does not clarify which statistical test was used.

The figures in the manuscript lack statistical analysis, and they do not include key statistical indicators such as p-values, standard deviations (SD), or error bars

Line 112 amikacin (AMK) was abbreviated on first use but for next time use, like at Line 142 use only abbreviation.

Comments on the Quality of English Language

Results section lacks an introductory sentence, making it abrupt. 

Antibiotic abbreviations should be consistently formatted and introduced upon first use

Author Response

Comment 1: I have a major issue with the RESULTS section of this manuscript as the authors JUMP directly to the conclusion without providing any context or introduction. This makes it difficult for readers to understand the significance of the results or how they relate to the study’s objectives 

Response 1: Thank you for your comment. We wanted to present the key findings in the first paragraph and then further elaborate on each finding in the discussion. However, we have now moved this paragraph to the end of the Discussion section providing a summarized idea of the discussion (page 8, paragraph 3, lines 272-279). 

Comment 2: Please describe what is Reincubation method in the introduction for general audience. 

 Response 2: We have described the Reincubation method in the Introduction section (page 3, paragraph 2, lines 112- 114).  

Comment 3: What is so special about these SLOMYCOI Sensititre™ microtiter plates? Please describe briefly for the journal audience. No explanation of why these plates were chosen, their specific composition, or their advantages. Please provide relevant information in introduction/result section for general audience. 

 Response 3: These plates, along with RAPMYCOI plates, are commonly used in the field of Nontuberculous Mycobacteria for susceptibility testing. We have included an explanation of the questions raised about these plates in the Introduction section ( page 3, paragraph 2, lines 114-119). 

Comment 4: Line 43- The manuscript states a 95.2% correlation but does not clarify which statistical test was used. The figures in the manuscript lack statistical analysis, and they do not include key statistical indicators such as p-values, standard deviations (SD), or error bars 

Response 4: We appreciate your comment on this topic. The term "correlation" was incorrectly used, as we intended to refer to the percentage of reproducibility of MBC values between the subculturing and Reincubation methods, rather than a statistical measure. We have corrected this in the manuscript and have included the formula used to calculate this value in the Methods section (page 9, paragraph 5, lines 328–331). 

Comment 5: Line 112 amikacin (AMK) was abbreviated on first use but for next time use, like at Line 142 use only abbreviation. 

Response 5: All abbreviations have been reviewed in the manuscript, including those in the figures and tables. 

Reviewer 3 Report

Comments and Suggestions for Authors

In this manuscript, the authors developed a new preincubation method to determine minimum bactericidal concentration (MBC) in nontuberculous mycobacteria (NTM). They determined the MBC values of 229 NTM clinical isolates by using the new method. However, there are some limitations that I believe should be improved.

Introduction:

1) The first paragraph should split into two parts between line 79-80.

2) Line 86: should “unoptimized” be “optimized”?

3) Line 87-89: what are other methods to determine MBC, beyond standard method the authors mentioned? This should be included in the introduction part.

Results:

1) How did authors calculate the percentage in line 110? It should be included in the methods part. What does correlation mean in line 110? Does it mean differences between two methods? What is the conclusion of Figure 1?

2) In Figure 2 and 3, what is the scale of y axis? What is shown in the bar graph? Median or mean? What is the standard deviation of MIC and MBC for each antibiotic?

Methods:

1) Line 263-265 and line 296-298 refers to five isolates. Does author use these five isolates in the corresponding result part?

Author Response

Comment 1: In this manuscript, the authors developed a new preincubation method to determine minimum bactericidal concentration (MBC) in nontuberculous mycobacteria (NTM). They determined the MBC values of 229 NTM clinical isolates by using the new method. However, there are some limitations that I believe should be improved. 

Introduction: 

1) The first paragraph should split into two parts between line 79-80.  

2) Line 86: should “unoptimized” be “optimized”?  

Response 1: We agree and are pleased to incorporate points 1 and 2 corrections into the manuscript. It has been thoroughly revised by a native English language expert before and after the resubmitted version of the manuscript. 

Comment 2: 3) Line 87-89: what are other methods to determine MBC, beyond standard method the authors mentioned? This should be included in the introduction part. 

Response 2: We have now included two sentences in the Introduction section (page 3, paragraph 2, lines 101-103, 123-125) mentioning other methods for MBC determination, along with a relevant citation [14-27]. 

Results: 

Comment 3: 1) How did authors calculate the percentage in line 110? It should be included in the methods part. What does correlation mean in line 110? Does it mean differences between two methods? What is the conclusion of Figure 1? 

Response 3: We appreciate your comment on this topic. The term "correlation" was incorrectly used, as we intended to refer to the percentage of reproducibility of MBC values between the subculturing and Reincubation methods, rather than a statistical measure. We have since corrected this in the manuscript and have included the formula used to calculate this value in the Methods section (page 9, paragraph 5, lines 328–331). 

Comment 4: Does it mean differences between two methods? 

Response 4: Yes, it does. The term refers to the difference in MBC values obtained using two different methods. However, to avoid confusion, we have revised the term to "reproducibility. 

Comment 5: What is the conclusion of Figure 1? 

Response 5: In Figure 1, we observed that almost all the antibiotics tested showed a variation of 0 or 1 dilution, with over 90% of MBC values matching between the two methods. Therefore, the conclusion is that the two methods are reproducible, as the MBC values are equivalent. This conclusion was already mentioned in the Results (page 4, paragraph 3, lines 133-136), Discussion (page 7, paragraph 2, lines 234-239; page 8, paragraph 3, lines 272-276) and Conclusion sections (page 10, paragraph 5, lines 373-375). 

Comment 6: 2) In Figure 2 and 3, what is the scale of y axis? What is shown in the bar graph? Median or mean? What is the standard deviation of MIC and MBC for each antibiotic? 

Response 6: The scale of the y axis is the concentration (MIC and the MBC) that kills 90% of the isolates (MIC90 and MBC90). The bar graph shows the values of the MIC90 and MBC90 for each species for each antibiotic tested as defined in the figure legend. It is not a median or mean value but a quartile 90 value of MIC and MBC. The concept of MIC90 and MBC90 is now explained in the Methods section (page 10, paragraph 4, lines 365-367) and in the legends of the figures. The MIC50 and MBC50 of each species is provided in Supplementary Materials. 

 Methods:  

Comment 7: Line 263-265 and line 296-298 refers to five isolates. Does author use these five isolates in the corresponding result part? 

Response 7: Thank you for your insight, it was typing error, which has now been corrected in both sections. 

Reviewer 4 Report

Comments and Suggestions for Authors

Based on the manuscript, here are the comments to be considered. 

  1. Why is there no reference reported in the method section for the standard method for MBC determination?
  2. Why did the author not compare the dilution-based MBC determination with the CFU based MBC determination?  
  3. Why did the author use only 7H9 as the growth media, as MAB and MAC require supplementation of either ADC, ADS, or OADC?
  4. How does it differ from existing reincubation methods? Author should included a comparison table highlighting the key differences among existing and new methods.
  5. What statistical methods were used to assess the correlation between the new reincubation method and the standard subculturing method for MBC determination? Please provide the correlation coefficient(s) and their statistical significance.
  6. The study mentions 229 NTM clinical isolates. Please provide details on the distribution of these isolates across different NTM species (e.g., M. avium, M. abscessus complex, M. xenopi, others). 
  7. Were any biases introduced by the selection of isolates?
    What quality control measures were implemented throughout the study to ensure the reproducibility and accuracy of the results?
  8. The conclusion states this information is important for optimizing combination therapy. However, the study doesn't explicitly propose specific optimized combination therapies based on the MBC/MIC data.
Comments on the Quality of English Language

Language is fine but can be improved. 

Author Response

Comment 1: Why is there no reference reported in the method section for the standard method for MBC determination? 

Response 1: Three references have been added to the Methods section [14,26,33] (page 9, paragraph 4, line 316), and 12 additional references have been included in the Introduction section [15-26] (page 3-4, paragraph 3, lines 123-125). 

Comment 2: Why did the author not compare the dilution-based MBC determination with the CFU based MBC determination? 

Response 2: This is an interesting question. We selected the dilution-based MBC method due to the extensive literature since 2013 supporting its use across various bacterial species [14-27], including mycobacteria [27]. Additionally, this comparison has already been comprehensively analyzed in a study by Hernandes et al. [14] and was therefore not conducted in our study. We adopted the subculturing method from their work, referred to as the 'flash microbicide method,' to compare it with the Reincubation method. A paragraph discussing the comparison between these two methods has been added to the Discussion section (page 7, paragraph 1, lines 215–221) and a comparison table (which you suggested adding) of these methods with the Reincubation method is now available in the Supplementary Materials section. 

Comment 3: Why did the author use only 7H9 as the growth media, as MAB and MAC require supplementation of either ADC, ADS, or OADC?  

Response 3: We have provided details of the medium used for each rapid and slow growing mycobacteria species in the Methods section (page 10, paragraph 2, lines 356-357, page 10, paragraph 3, lines 360-361) as some NTM species have specific nutrition requirements.  

 Comment 4: How does it differ from existing reincubation methods? Author should include a comparison table highlighting the key differences among existing and new methods.  

Response 4: We have created a simplified table highlighting the key differences between the existing and the new methods. However, significant variations in these methods have been reported [11]. The most common sources of variability include the type of culture media used, incubation time, and the percentage of bacterial death required for a drug to be considered bactericidal. To address this, we have included a table in Supplementary Material (Table S3) that compares the two most used methods for determining MBC with the Reincubation method. 

Table S3: Comparison of commonly used minimum bactericidal concentration determination methods with the method used by our group. 

Method 

Description 

Advantages 

Disadvantages 

Reincubation 

MIC plate is reincubated and the MBC is determined by visual bacterial growth as + or -. 

  • Simplified protocol 
  • Faster results 
  • Low workload 

Broth  Microdilution or broth Macrodilution + CFU counting onto agar [11] 

Content from MIC plates or tubes is transferred onto agar plates and CFUs are counted. 

  • Provides quantitative data on the MBC. 
  • Widely accepted 
  • Clear results 
  • Clear endpoints 
  • Reliable  
  • Labor intensive 
  • Time consuming  

Broth Microdilution + liquid subculture (+/-) [14-27] 

Content from MIC plates is transferred to fresh liquid medium and MBC is determined by  visual bacterial growth as + or -. 

  • Provides quantitative data on the MBC.  
  • Widely accepted 
  • Clear results 
  • Clear endpoints 
  • Reliable 
  • Labor intensive 
  • Time consuming  

Comment 5: What statistical methods were used to assess the correlation between the new reincubation method and the standard subculturing method for MBC determination? Please provide the correlation coefficient(s) and their statistical significance. 

Response 5: The term "correlation" was mistakenly used as we were referring to the percentage of reproducibility of MBC values between the subculturing and Reincubation methods, rather than a statistical measure. We have now corrected this in the manuscript and have also included the formula used to calculate this value in the Methods section (page 10, paragraph 5, lines 328–331). 

Comment 6: The study mentions 229 NTM clinical isolates. Please provide details on the distribution of these isolates across different NTM species (e.g., M. avium, M. abscessus complex, M. xenopi, others).  

Response 6: The distribution of the isolates is well defined in Method section (page 9, paragraph 6, lines 337-341).   

Comment 7: Were any biases introduced by the selection of isolates? 
What quality control measures were implemented throughout the study to ensure the reproducibility and accuracy of the results? 

Response 7: The “isolate selection” section for the study of the 229 NTM clinical isolates has been modified in the manuscript to “isolate recruitment” for accuracy, as they were isolates consecutively recruited during the 18 months that the study was carried out. The criteria to be selected for isolate recruitment has been added to the Methods section (page 10, paragraph 1, lines 341-344). The reproducibility and accuracy of the results was ensured by performing the experiments in duplicate (page 9 paragraph 1, line 298; page 9 paragraph 3, line 314; page 9, paragraph 4, line 322, page 9 paragraph 5, line 334; page 10, paragraph 5, line 370). 

Comment 8: The conclusion states this information is important for optimizing combination therapy. However, the study doesn't explicitly propose specific optimized combination therapies based on the MBC/MIC data. 

Response 8: The aim of our study was not to propose a specific antibiotic combination but to identify the most effective antibiotics for use in combination based on their bactericidal activity. We completely agree that the information provided in the conclusion was speculative rather than conclusive, and thus, we have moved this point to the Discussion section (page 7, paragraph 1, lines 244-245) and deleted it from the Conclusion section. 

Round 2

Reviewer 3 Report

Comments and Suggestions for Authors

The revised manuscript has been sufficiently improved to warrant publication in Antibiotics.

Author Response

Comment 1: The revised manuscript has been sufficiently improved to warrant publication in Antibiotics. 

Response 1: Thank you for your positive assessment of our revised manuscript. We appreciate your time and effort in reviewing our work and providing valuable feedback, which has helped us improve the quality of our study. We are grateful for your support and consideration for publication in Antibiotics. 

Reviewer 4 Report

Comments and Suggestions for Authors

The manuscript is significantly improved, and the authors answered all queries. 

The major point here is if the reproducibility of MBC of ETH is 76% and RIF is not a suitable candidate for this method, how author can generalize this method to screen MIC/MBC of novel drug candidates? 

Author should comment on: What may the potential factors specific to certain compounds/drugs which can make this method ineffective. 

Author Response

Comment 1: The major point here is if the reproducibility of MBC of ETH is 76% and RIF is not a suitable candidate for this method, how author can generalize this method to screen MIC/MBC of novel drug candidates?

Response 1: Thank you for your insightful comment. The main issue with rifampicin seems to be the degradation due to long incubation, which is affected in MBC determination, preventing us from determining the MBC using our intended methodology. Our group is currently conducting a study on rifampicin degradation. 

ETH demonstrated 76% reproducibility, which is not ideal but still acceptable, while the reproducibility for the other tested antibiotics exceeded 90%. Based on these results, we believe this method is reliable for the tested antibiotics. However, for other or novel drug candidates, a comparative study of both methods should be conducted beforehand. 

Comment 2: Author should comment on: What may the potential factors specific to certain compounds/drugs which can make this method ineffective. 

Response 2: Many potential compound-specific factors such as solubility, stability, and interaction with media components would make the MBC determination method ineffective and therefore a comparative study of both methods should be conducted beforehand. We have added these lines in the discussion section to address this comment (page 7, 4 last lines in paragraph 1): “Compound-specific factors such as solubility, stability, and interaction with media components might also affect the Reincubation method for the MBC determination. Therefore, prior MBC testing by comparing Reincubation and subculturing methods should be conducted in other or novel drugs that have not been tested in this study.